**Data Availability Statement:** All relevant data are within the paper and its Supporting Information

# Development of a multivariate predictive model for dapsone adverse drug events in people with leprosy under standard WHO multidrug therapy

Ana Carolina Galvão dos Santos de Araujo[1]*, Mariana de Andrea Vilas-Boas Hacker[2], Roberta Olmo Pinheiro[2], Ximena Illarramendi[2], Sandra Maria Barbosa Durães[3], Maurício Lisboa Nobre[4], Milton Ozório Moraes[2†], Anna Maria Sales[2☉], Gilberto Marcelo Sperandio da Silva[1☉]

**1** Evandro Chagas National Institute of Infectious Diseases, Oswaldo Cruz Foundation (INI/Fiocruz), Rio de Janeiro, RJ, Brazil, **2** Oswaldo Cruz Institute, Oswaldo Cruz Foundation (IOC/Fiocruz), Rio de Janeiro, RJ, Brazil, **3** Dermatology Department, Fluminense Federal University (UFF), Niterói, RJ, Brazil, **4** Giselda Trigueiro Hospital, Rio Grande do Norte Federal State Public Health department (SESAP-RN), Natal, RN, Brazil

☉ These authors contributed equally to this work.
† Deceased.
* anacarolinagsa@yahoo.com.br

## Abstract

### Background

The occurrence of adverse drug events (ADEs) during dapsone (DDS) treatment in patients with leprosy can constitute a significant barrier to the successful completion of the standardized therapeutic regimen for this disease. Well-known DDS-ADEs are hemolytic anemia, methemoglobinemia, hepatotoxicity, agranulocytosis, and hypersensitivity reactions. Identifying risk factors for ADEs before starting World Health Organization recommended standard multidrug therapy (WHO/MDT) can guide therapeutic planning for the patient. The objective of this study was to develop a predictive model for DDS-ADEs in patients with leprosy receiving standard WHO/MDT.

### Methodology

This is a case-control study that involved the review of medical records of adult (≥18 years) patients registered at a Leprosy Reference Center in Rio de Janeiro, Brazil. The cohort included individuals that received standard WHO/MDT between January 2000 to December 2021. A prediction nomogram was developed by means of multivariable logistic regression (LR) using variables. The Hosmer–Lemeshow test was used to determine the model fit. Odds ratios (ORs) and their respective 95% confidence intervals (CIs) were estimated. The predictive ability of the LRM was assessed by the area under the receiver operating characteristic curve (AUC).

files. Our institution complies with open science and open data policies, and we agreed to make available the data used for this study that was obtained from the patients of our center. We have deposited the data and metadata at our institution repository, doi:10.35078/X3HKNS.

**Funding:** The work was financially supported by CNPq - Conselho Nacional de Desenvolvimento Científico e Tecnológico (URL: https://www.gov.br/cnpq/pt-br, Grant number: 312802/2020-0 to ROP), FAPERJ - Fundação Carlos Chagas Filho de Amparo à Pesquisa do Estado do Rio de Janeiro (URL: https://www.faperj.br/, Grant number: E-26/201.176/2021 (260734) to ROP), and FIOCRUZ - Fundação Oswaldo Cruz. The funders had no role in study design, data collection and analysis, decision to publish, or preparation of the manuscript.

**Competing interests:** The authors have declared that no competing interests exist.

## Results

A total of 329 medical records were assessed, comprising 120 cases and 209 controls. Based on the final LRM analysis, female sex (OR = 3.61; 95% CI: 2.03–6.59), multibacillary classification (OR = 2.5; 95% CI: 1.39–4.66), and higher education level (completed primary education) (OR = 1.97; 95% CI: 1.14–3.47) were considered factors to predict ADEs that caused standard WHO/MDT discontinuation. The prediction model developed had an AUC of 0.7208, that is 72% capable of predicting DDS-ADEs.

## Conclusion

We propose a clinical model that could become a helpful tool for physicians in predicting ADEs in DDS-treated leprosy patients.

### Author summary

Adverse events (AE) produced by the drugs used to treat leprosy can hinder the successful completion of the therapeutic regimen. Well-known AE produced by dapsone (DDS) are related to liver problems, allergic reactions, or to the destruction of red and/or white blood cells, causing anemia. Helping the physician to recognize a patient that may develop these adverse reactions can be useful. Thus, we developed a model to predict AE in patients with leprosy receiving standard World Health Organization-recommended multidrug therapy (WHO/MDT). Our question was whether we could use sociodemographic and clinical variables to generate a predictive model for DDS-ADEs. The model developed in this study could be a useful tool to assist physicians in predicting DDS-ADEs when treating patients with standard WHO/MDT for leprosy, and thus, establish a safer therapeutic plan for patients with a greater ADE risk.

## Introduction

The occurrence of adverse drug events (ADEs) during dapsone (DDS) treatment in patients with leprosy can constitute a significant barrier to the successful completion of the standardized therapeutic regimen for this disease. Currently, DDS, together with rifampicin and clofazimine, comprise the standard multidrug therapy (MDT) recommended by the World Health Organization (WHO) to treat leprosy [1,2].

DDS is a drug of the sulfone class, which was synthesized in 1908 and first used as an antimicrobial in 1937, to treat streptococcal infections in rats. In 1943, DDS was revealed as a treatment for leprosy, changing the history of the disease [3–5]. DDS is a bacteriostatic antibiotic that exerts its effects by inhibiting dihydrofolic acid synthesis. Furthermore, it may suppress the expression of inflammatory signaling pathways and the generation of reactive oxygen species (ROS) [6,7].

DDS can cause dapsone hypersensitivity syndrome (DHS), which has an estimated prevalence of 0.5–3.6% and fatality rate of 9.9% [8,9]. DHS is characterized by fever, rash, lymphadenopathy, and hepatitis, which usually develops after patients receive DDS for 5 to 6 weeks [10]. DDS has also been associated with hemolytic anemia, methemoglobinemia, hepatotoxicity, agranulocytosis, and other severe cutaneous adverse reactions (SCARs), such as Stevens–

Johnson syndrome (SJS) and toxic epidermal necrolysis (TEN), which are associated with mortality rates of up to 5% and 30%, respectively [11,12].

While DHS is restricted to HLA-B*13:01, the positive predictive value of this allele is only 7.8% [8]. To explore the potential coexisting factors involved in the occurrence of DHS, a genomewide association study (GWAS) together with genomewide DNA methylation profile analysis comparing patients with DHS and DDS-tolerant control patients was performed. Sun et al. (2023) [13] demonstrated differences in immune responses between patients with DHS and the DDS-tolerant controls. The authors found that the ability of antigen-presenting cells to activate DDS-specific T cells was enhanced in patients with DHS compared with those of DDS-tolerant controls. Although epigenetic regulation may be associated with hypersensitivity, clinical parameters, and sociodemographic factors should also be considered in the development of a predictive model for ADEs [13,14]. For example, delayed hemolytic anemia was seen in patients with high methemoglobin levels during presentation [15].

Patient compliance is a multifactorial problem with marked consequences for the success of disease control policies in endemic areas. Low compliance to standard MDT may have serious implications for the leprosy control program because it can set the stage for the emergence of drug resistance, eventually resulting in treatment failure and failure of the program altogether. Besides morbidity, MDT ADEs, especially those that are DDS-related (DDS-ADEs), can cause nonadherence of treatment. Previous studies in Brazil, associated ADEs with treatment nonadherence in 14.9% [16] and 24% [17] of patients treated with standard MDT. In the Philippines, ADEs were indicated by the patients as the most important reason for interrupting treatment [18].

Changes in the scheme and length of leprosy treatment are still being studied, and recently, in 2021, clofazimine was included in the paucibacillary (PB) treatment scheme [19]; however, no other regimen has been demonstrated to be better than the standard WHO/MDT. DDS continues to be used mainly owing to its immunomodulatory action [20]. Therefore, identifying patients with risk factors for ADEs, before starting MDT, can help design a safer therapeutic plan for the patient under higher risk.

Mathematical models have gained widespread use in predicting disease prognosis and treatment-related adverse reactions. The nomogram model is the most effective visualization tool for regression equations. Nomograms allow integration and synthesis of the relative importance of clinical variables and provide a graphical representation of the odds ratios (ORs), p-values, and confidence intervals (CI) of logistic regression models(LRM), enabling prediction of the probability of event occurrence. The objective of this study was to identify patients with a higher sensitivity to develop DDS-ADEs during the treatment with standard WHO/MDT. For this, we developed a predictive model for DDS-ADEs in patients with leprosy under standard WHO/MDT. Using this tool to predict DDS-ADEs based on sociodemographic and clinical variables could be an inexpensive way to decrease MDT nonadherence and ADE morbidity.

## Methods

### Ethics statement

The study was approved by the Institutional Review Board of the Oswaldo Cruz Institute (IOC/Fiocruz) (CAAE: 54746021.2.0000.5248, approval number: 5.290.811). Informed consent was waived because of the retrospective nature of the study.

## Study design

This is a case-control study, nested in a cohort of individuals treated for leprosy from January 2000 to December 2021 at the Souza Araujo outpatient clinic (ASA) of the Oswaldo Cruz Institute (IOC), Oswaldo Cruz Foundation (FIOCRUZ), Rio de Janeiro, Brazil. ASA is a Ministry of Health referral center for diagnosing and treating people affected by leprosy. Patients who started and completed leprosy treatment at the ASA during the aforementioned observation period were included in the study. Patients who had insufficient clinical data for ADE causality classification, such as detailed symptom descriptions or physical examination findings, were excluded.

The cases comprised individuals who needed to interrupt standard MDT owing to DDS-ADEs, and subsequently started alternative MDT.The controls comprised individuals who completed standard WHO/MDT without drug intercurrences. According to the ADE severity classification by Yun et al. (2009), all events in this study were considered severe as they required drug interruption [21].

The standard PB therapeutic regimen was daily DDS with monthly rifampicin for 6 months; while the standard multibacillary (MB) therapeutic regimen was daily DDS and clofazimine with monthly rifampicin plus a higher dose of clofazimine for 12 months.

The medical records of patients included in the study were reviewed for age, sex, self-referred skin color, leprosy clinical form, symptoms, and hematology and biochemistry laboratory results when available (before treatment and at ADE onset). For individuals with medication intolerance during MDT, the timing of treatment suspension was recorded. Information about glucose-6-phosphate dehydrogenase (G6PD) status, which is linked to hemolytic anemia, was not available. All data were collected and managed using the software Research Electronic Data Capture (REDCap) tools hosted at Evandro Chagas National Institute of Infectious Diseases.

Causality of ADEs was determined according to the Naranjo scale [22]. The main ADEs were classified as described below. Anemia was considered if the hemoglobin values were less than 13 g/dL in males or 11 g/dL in females, or if the hematocrit level was lower than 40% in males or 34% in females, and/or symptoms were reported that could be consistent with symptomatic anemia. Methemoglobinemia was considered separately, and the diagnosis was based on the presence of cyanosis, and/or dyspnea, without changes in the hemoglobin and hematocrit levels. Gastrointestinal side effects included anorexia, nausea, vomiting, diarrhea, and gastric or epigastric pain. Dermatological side effects included rash and/or pruritus. DHS was diagnosed according to Richardus and Smith's (1989) classification: the presence of two or more criteria among fever, skin rash, lymphadenopathy, hepatotoxicity (hepatomegaly, jaundice, or laboratory enzyme alteration), 2–8 weeks after beginning treatment with DDS, with regression after stopping treatment in cases where the symptoms were not associated with other diseases, other medications, or leprosy reactions [10]. The presence of DDS-ADEs (Naranjo's classification as possible or probable) was considered the primary outcome of the present study.

For continuous variables, descriptive statistics were performed using mean values with the standard deviation, and a Student's T-test was used to compare data between groups. Categorical variables were described using frequency (N) and proportions (%). The Pearson's Chi-squared test was used to compare categorical variables between groups (with and without ADEs).

LRM was performed to estimate the probability of developing ADEs that caused MDT to be discontinued owing to DDS. All variables for which the p-value in univariate analysis was lower than 0.20 were included in a stepwise final LRM. This was used to develop a

multivariable prediction nomogram. The Hosmer–Lemeshow test was used to show how adequately the model fits the data. ORs and their respective 95% CIs were also estimated. The predictive ability of the LRM was assessed by the area under receiver operating characteristic (ROC) curve (AUC).

P-values of 0.05 or less were considered statistically significant. Statistical analyses and data visualization were performed using RStudio.

## Results

Data from 433 patients with leprosy registered at the ASA Database were assessed, and cases that received standard MDT during the observation period were initially selected (Fig 1). After exclusion criteria were applied, a total of 329 medical records were included for analysis in the study. Of these, 120 were from patients that had DDS-ADEs (cases), while 209 had no registry of ADEs (controls).

Table 1 shows the main characteristics of the case and control groups. Females comprised 65.29% (79/120) of the case group and 35.89% (75/209) of the control group (p<0.001). The

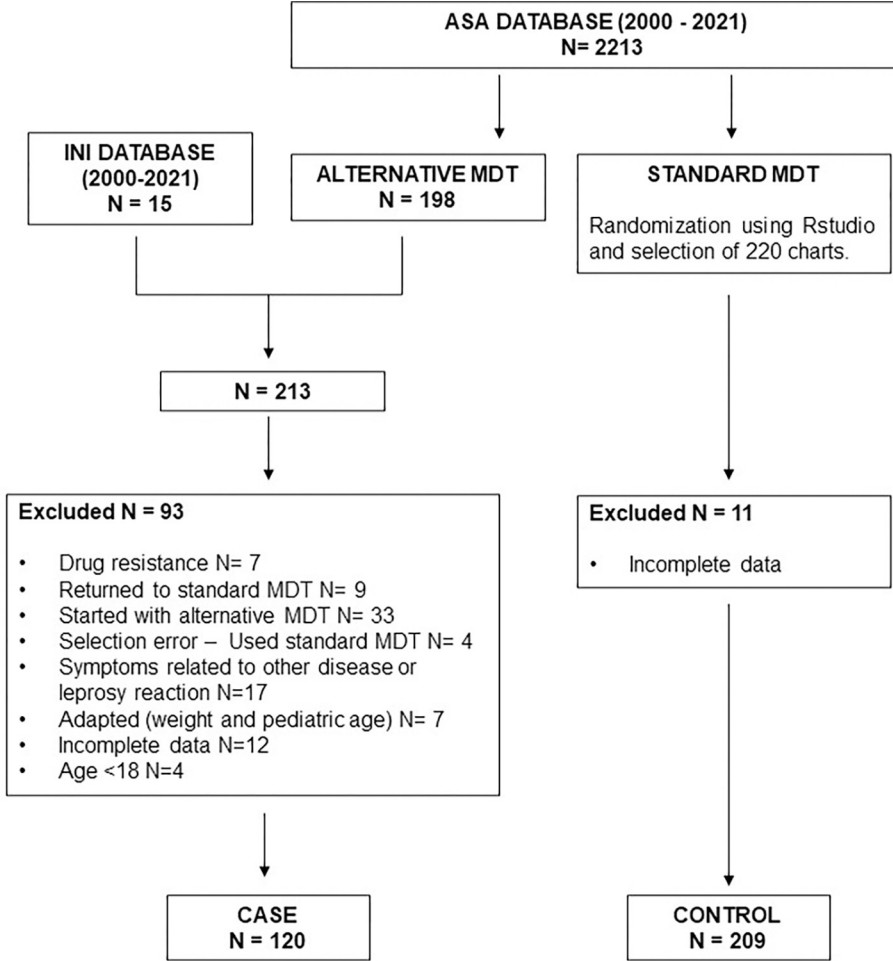

**Fig 1. Flowchart of the study.** ASA, Souza Araujo outpatient clinic; INI, Evandro Chagas National Institute of Infectious Diseases; MDT, multidrug therapy. Incomplete data included patients records without information about symptoms during follow-up.

**Table 1. Sociodemographic, clinical and laboratorial characteristics of 329 patients with leprosy who were treated at the Souza Araujo outpatient clinic (ASA), Rio de Janeiro, Brazil, between January 2000 and December 2021.**

| | | CASES 120 N(%) | CONTROLS 209 N(%) | p-value |
|---|---|---|---|---|
| **Sex** | Female | 79 (65.83%) | 75 (35.89%) | <0.001[a] |
| | Male | 41 (34.17%) | 134 (64.11%) | |
| **Age (years)** | Mean (±SD) | 49.2 (17) | 45.9 (15.3) | 0.073[b] |
| | 18–47 | 52 (43.3%) | 115 (55.02%) | 0.051[a] |
| | >47 | 68 (56.7%) | 94 (44.98%) | |
| **Self-referred skin color** | White | 72 (60%) | 107 (55.02%) | 0.153[a] |
| | Non-white | 44 (40%) | 94 (44.98%) | |
| **Income (minimum wages)** | 0–2 | 67 (55.83%) | 112 (53.59%) | 0.632[a] |
| | >2 | 54 (45%) | 102 (48.80%) | |
| **Education** | Incomplete primary education | 56 (48.28%) | 107 (55.44%) | 0.222[a] |
| | Completed primary education | 60 (51.72%) | 86 (44.56%) | |
| **Marital status** | Not married | 60 (53.1%) | 114 (58.8%) | 0.397[a] |
| | Married | 53 (46.9%) | 80 (41.2%) | |
| **Operational classification for treatment** | MB | 58 (48.33%) | 110 (52.63%) | 0.453[a] |
| | PB | 62 (51.67%) | 99 (47.37%) | |
| **Number of skin lesions at diagnosis** | 0–5 | 65 (54.17%) | 89 (42.58%) | 0.195[a] |
| | >5 | 46 (38.33%) | 89 (42.58%) | |
| **Leprosy reaction at diagnosis** | Yes | 12 (10%) | 36 (17.22%) | 0.0216[a] |
| | No | 108 (90%) | 137 (65.55%) | |
| **Bacilloscopic index at diagnosis** | 0 | 65 (54.17%) | 105 (50.24%) | 0.498[a] |
| | >0 | 54 (45%) | 102 (48.80%) | |
| **Gastrointestinal disease** | Yes | 15 (12.5%) | 12 (5.74%) | 0.039[a] |
| | No | 103 (85.8%) | 187 (89.47%) | |

[a] Pearson's Chi-squared test

[b] Student's T-test. MB, multibacillary; PB, paucibacillary.

mean age (±SD) was 49.2 (17) years in the case group and 45.9 (15.3) in the control group; however, this difference was not significant (p = 0.073). When age was categorized, older adults (over 47 years old) were 1.6 times more affected by ADEs (p = 0.051; Table 1).

We compared the clinical manifestation of leprosy between the case and control groups using the bacilloscopic index, number of lesions, and MB versus PB leprosy and no statistically significant differences were observed.

Patients with gastrointestinal diseases had a 2.27 times higher risk of ADEs (p = 0.039; Table 1 and 2), while no statistically significant differences were seen between the groups regarding the other comorbidities considered (S1 Table). Furthermore, patients diagnosed with leprosy reactions at diagnosis and the beginning of MDT had fewer ADEs (p = 0.0216; Tables 1 and 2).

To assess for confounding between the variables, multiple logistic regression analyses were completed. Female sex (OR = 2.66 CI:(0.22–0.66), p = <0.001) remained associated with higher chance of DDS intolerance, and MB leprosy showed a higher chance of ADEs Table 2.

Regarding the most frequent ADEs, anemia was the most common, with 75 (62.5%) patients experiencing this, followed by DHS (16 patients, 13.22%). Other frequent ADEs were methemoglobinemia, gastrointestinal intolerance, and other drug eruptions (Table 3).

Sex, age, gastrointestinal disease, and leprosy reaction at diagnosis were included in the first regression model owing to their association with the occurrence of overall ADEs, while self-referred skin color, education, and number of skin lesions at diagnosis were included due to a

**Table 2. Univariate and multivariate analysis of adverse drug events associated with dapsone, using sociodemographic and clinical variables of patients with leprosy treated with standard MDT.**

| Variables | | Univariate analysis ORc (95% CI) | Multivariate analysis ORa (95% CI) |
|---|---|---|---|
| **Sociodemographic** | | | |
| **Sex** | Female | 3.4 (2.1–5.5) | 3.8 (1.98–7.59) |
| | Male | 1 | |
| **Age (years)** | 18–47 | 1 | |
| | >47 | 1.6 (1.01–2.51) | 1.26 (0.68–2.34) |
| **Self-referred skin color** | Non-white | 1 | |
| | White | 1.43 (0.9–2.3) | 1.77 (0.95–3.33) |
| **Income (minimum wages)** | 0–2 | 1 | |
| | >2 | 0.89 (0.56–1.43) | 1.29 (0.66–2.56) |
| **Education** | Incomplete primary education | 1 | |
| | Completed primary education | 1.33 (0.84–2.12) | 1.68 (0.86–3.33) |
| **Marital status** | Not married | 1 | |
| | Married | 1.26 (0.79–2.01) | 0.94 (0.5–1.75) |
| **Clinical variables** | | | |
| **Operational classification for treatment** | PB | 1 | |
| | MB | 0.84 (0.54–1.32) | 3.81 (1.47–10.98) |
| **Number of skin lesions at diagnosis** | 0–5 | 1 | |
| | >5 | 0.71 (0.44–1.14) | 0.64 (0.23–1.69) |
| **Leprosy reaction at diagnosis** | No | 1 | |
| | Yes | 0.43 (0.20–0.84) | 0.46 (0.17–1.18) |
| **Disability grade at diagnosis*** | 0 | 1 | |
| | >0 | 1.07 (0.67–1.7) | 1.18 (0.61–2.28) |
| **Gastrointestinal disease** | No | 1 | |
| | Yes | 2.27 (1.01–5.26) | 2.36 (0.81–7.54) |

a OR, adjusted odds ratio; c OR, Crude odds ratio; CI, confidence interval; MB, multibacillary; PB, paucibacillary.

*Disability grade classifies patients according to sensory and motor neurological impairment and deformity of face, hand(s) and feet; the lowest value being 0, no disability, and the highest = 2, permanent disability and/or deformity.

**Table 3. Adverse drug events associated with dapsone among 120 patients with leprosy who were treated at the Souza Araujo outpatient clinic (ASA), Rio de Janeiro, Brazil, between January 2000 and December 2021.**

| Adverse drug event | N = 120 N(%) |
|---|---|
| **Anemia** | 75 (62.5%) |
| **Dapsone hypersensitivity syndrome** | 16 (13.33%) |
| **Methemoglobinemia** | 9 (7.5%) |
| **Gastrointestinal intolerance** | 7 (5.83%) |
| **Drug eruption** | 4 (3.33%) |
| **Myalgia and malaise** | 3 (2.5%) |
| **Agranulocytosis** | 1 (0.83%) |
| **Asthenia** | 1 (0.83%) |
| **Dizziness** | 1 (0.83%) |
| **Hepatotoxicity** | 1 (0.83%) |
| **Isolated malaise** | 1 (0.83%) |
| **Pancytopenia** | 1 (0.83%) |

**Table 4. Final logistic regression model after backward elimination.**

| Variables | | p-value | OR | 95% CI |
|---|---|---|---|---|
| **Sex** | Male<br>Female | <0.001 | 1<br>3.61 | (2.03–6.59) |
| **Operational classification for treatment** | PB<br>MB | 0.003 | 1<br>2.5 | (1.39–4.66) |
| **Education** | Incomplete primary education<br>Completed primary education | 0.02 | 1<br>1.97 | (1.14–3.47) |
| **Gastrointestinal disease** | No<br>Yes | 0.057 | 1<br>2.68 | (1–7.84) |
| **Leprosy reaction at diagnosis** | No<br>Yes | 0.082 | 1<br>0.48 | (0.2–1.08) |
| **Self-referred skin color** | Non-white<br>White | 0.253 | 1<br>1.39 | (0.79–2.43) |
| **Age (years)** | 18–47<br>>47 | 0.335 | 1<br>1.32 | (0.75–2.31) |

OR, odds ratio; CI, confidence interval; MB, multibacillary; PB, paucibacillary.

p-value lower than 0.2. Income, marital status, disability grade, and operational classification for treatment were included because of their clinical relevance [23].

A new model was calculated excluding the variables with a p-value >0.2 but maintaining the inclusion of important sociodemographic and clinical characteristics, such as age, education level, and operational classification for treatment. A goodness-of-fit test demonstrated a good performance of the model (p = 0.86). Based on the final model results, given in Table 4, the following variables were considered to predict overall ADEs: female sex (OR, 3.61; 95% CI, 2.03–6.59), MB leprosy (OR, 2.5; 95% CI, 1.39–4.66), and completion of primary education (OR, 1.97; 95% CI 1.14–3.47).

This model was used to develop a nomogram to determine the risk of DDS-ADEs in patients with leprosy (Fig 2).

The area under the ROC curve (AUC), which assesses the ability of the LRM to predict overall ADEs, was 0.72 (95% CI, 0.66–0.78) (Fig 3).

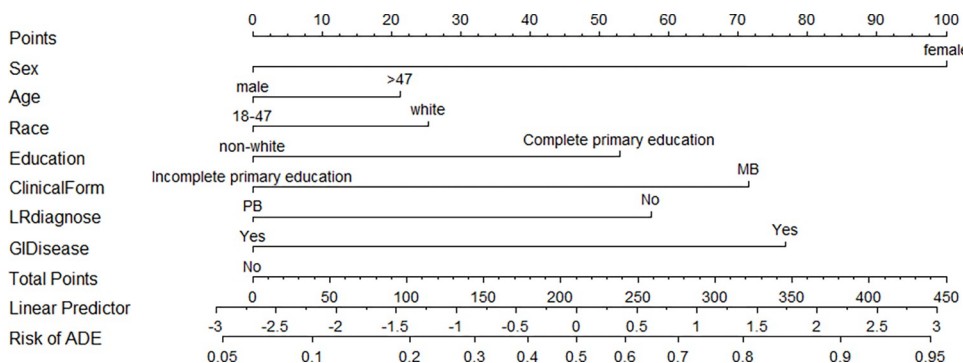

**Fig 2. Nomogram to identify the risk of dapsone-related adverse drug events (DDS-ADEs) in patients with leprosy, based on logistic regression analysis.** A vertical line directed to the "Points" axis is used to acquire the corresponding scores for each variable. After adding each variable value, a line is drawn from the "Total Points" axis to the "Risk of ADE" axis to determine the risk of an ADE to DDS. ClinicalForm, operational classification for treatment; GIDisease, gastrointestinal disease; LRdiagnose, Leprosy reaction at diagnosis; MB, multibacillary; PB, paucibacillary.

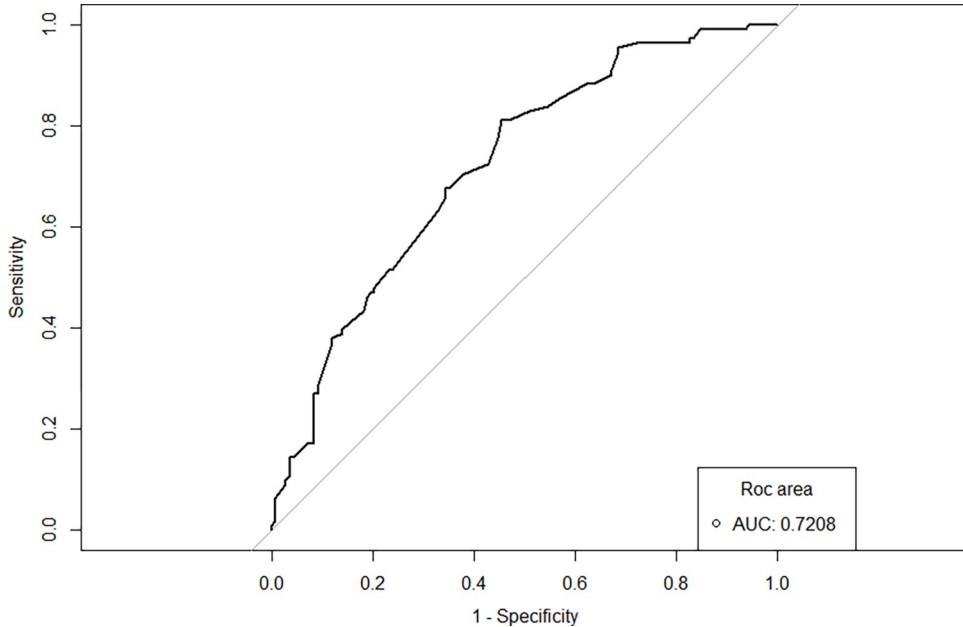

**Fig 3. The receiver operating characteristic (ROC) curve was based on the probability of development of dapsone-related adverse drug events (DDS-ADEs), calculated by the final logistic regression model.** AUC, area under the ROC curve.

## Discussion

ADEs can lead to higher morbidity during leprosy treatment, and DDS is the primary drug that leads to interruption of standard MDT treatment owing to the occurrence of ADEs [11]. Our study selected patients who had severe DDS-ADEs and needed to stop standard MDT. This explains the higher frequency of DHS, an ADE that is well known for its high morbidity, than other more common but better tolerated events, such as gastrointestinal symptoms [11,24].

In this study, DDS-ADEs were more frequently observed among female patients. This finding is consistent with other studies that have reported a higher frequency of DDS-ADEs in this sex [11,25]. Although studies have previously demonstrated that DDS-ADEs occur almost twice as often in females than in males, the role of sex as a biological factor in the generation of ADEs is still poorly understood [25].

General differences in body weight and the percentage of body fat between the sexes can affect the absorption and distribution of drugs. Unfortunately, we did not have information regarding weight and height in the database, to include this variable in the model. We cannot exclude the fact that several biological, psychological, and cultural factors may contribute for differences in the ratio of ADEs between the sexes, including pharmacokinetics (PK) and pharmacodynamics (PD) [26,27]. One additional hypothesis for the higher ratio of DDS-ADEs in females in the present study is that blood loss during the menstrual period could play a role in anemia predisposition; however, other hormonal effects could also be involved [25].

Previous studies have demonstrated that older age groups are more likely to experience ADEs [28]. In the present study, we demonstrated that the older participants (>47 years) had more ADEs than younger ones. Goulart et al. (2002) and Tortelly et al. (2021) also found a higher risk of ADEs in older adults, contrary to the findings by Dupnik et al. (2013), who reported a higher risk of DDS-ADEs in younger women [11,16,25]. One limitation of our

study was that we could not measure the impact of polypharmacy in the ADE risk since we did not have the overall relation of drugs used by the patients during the study, although polypharmacy has also been linked to ADEs in older people. PK and PD aspects in older adults could also impact the occurrence of ADEs.

DDS is absorbed slowly after oral administration. Peak plasma drug concentration is reached in approximately 4 h, absorption half-life is 1.1 h, and elimination half-life is approximately 30 h. Oral availability is around 90%. DDS is metabolized via acetylation and N-hydroxylation, but acetylation polymorphism has no effect on DDS handling [29]. However, the rate of acetylation depends on the acetylator phenotypes [30]. During MDT, the potent antibiotic rifampicin induces the metabolism of DDS, which results in a decreased plasma half-life of DDS and its metabolites [31]; however, no differences in DDS PK and PD related to sex or age have been reported in patients with leprosy. Future studies from our group will evaluate DDS and its metabolites, monoacetyl dapsone (MADDS) and diacetyl dapsone (DADDS), in plasma and urine from patients with leprosy.

We found that leprosy patients treated with MB MDT were associated with DDS-ADEs. Kubota et al. (2014) observed that, among patients who underwent alternative MDT, ADEs were more frequent in patients with MB leprosy. Goulart et al. (2002) did a chart review of 187 patients treated between 1995 and 2000 and also found a higher frequency of ADEs in patients with MB leprosy [16,23]. Generally, the symptoms begin in the first 6 doses of MDT, and thus, the predisposition of patients with MB leprosy cannot be explained by the longer treatment duration. In our study, only 4 (3.33%) of the ADEs happened after the sixth dose of MDT, and 100 out of 120 (83.33%) ADEs happened until the third dose [11,16].

Gallo et al. (1995) observed a similar proportion of DDS ADE in patients treated with PB and MB MDT, while Dupnik et al. (2013) showed that patients with PB leprosy had more ADEs than those treated with MB MDT. However, Dupnik et al. questioned whether the higher frequency of females in the PB group could interfere with this observation. In our study, patients with MB leprosy had a greater chance of developing ADEs, despite a minority of females in this group (33%), in comparison to the patients with PB leprosy where 60% were females. [14,25].

Regarding educational status, higher educational levels were associated with the presence of ADEs. This association has not been reported in other leprosy studies; however, higher educational levels could lead to a better understanding of the occurrence of drug-related symptoms and, thus, increase the probability of a patient reporting the symptoms, as was described recently by Costa et al. (2023) [28]. Other studies have shown that higher educational levels in different diseases could help medication intake and reduce ADEs [32,33].

Leprosy reactions at diagnosis and the beginning of treatment were a protection factor for ADEs, although the difference between the case and control groups was not statistically significant. Our theory is that systemic corticosteroids and, sometimes, thalidomide are used in the treatment of these reactions and, as both drugs have anti-inflammatory properties, they may mask ADE symptoms [34].

Gastrointestinal disease was associated with a higher frequency of ADEs, although this difference was not significant. Unfortunately, another limitation of this study is that it was not possible to classify the different gastrointestinal disease diagnoses. Patients with gastrointestinal diseases may have chronic mucosal bleeding, malabsorption, and diarrhea, which can lead to increased susceptibility to anemia, dehydration, loss of electrolytes, and less absorption of iron and other vitamins. Inflammation may also play a role in iron homeostasis. In this context, patients with gastrointestinal disease can become more susceptible to ADEs [35].

Despite a lack of statistical significance, self-referred skin color was included in the model since it is known that genetics play an important role in ADE prediction. For example, it is

known that G6PD deficiency is associated with hemolytic anemia and that the presence of HLA-B13*01 is associated with DDS-related SCARs [36,37]. Laboratory data was missing for many patients, which was a limitation of this study. Inclusion of hemoglobin and G6PD levels, pharmacogenomic data, or other laboratory test results may increase the accuracy of the model.

After backward elimination of the non significant variables of the first LRM, all those variables with a p-value <0.2 were selected to compose the final model. The result was presented as a prediction model nomogram that had an AUC of 0.7208. The AUC assesses the ability of the LRM to predict ADEs that caused DDS-ADEs. In other words, this model is 72% capable of predicting DDS-ADEs.

Although the prediction nomogram allows the identification of patients that need a close monitoring, the present study has some limitations, that include the retrospective design, the absence of laboratory tests and body weight of the patients. This model reflects the outcomes of the regression analyses and further refinement is recommended to enhance its practical applicability in clinical settings. To achieve this, both internal and external validation must be conducted. Hilder and Lockwood (2020), in their systematic review, recommended that patients should have laboratory examinations when starting MDT with repeated assessments at 4–8-week intervals. In resource-constrained environments, particularly in low and middle-income countries, this model could provide guidance to identify patients in need of more frequent monitoring [38].

## Conclusion

Identifying sociodemographic and clinical characteristics, such as sex, age, self-referred skin color, education status, clinical form of disease manifestation, and comorbidities that can predict ADEs can allow the establishment of strategies to minimize MDT complications. Although the nomogram cannot substitute the clinical judgment, the prediction model developed in this study has an AUC of 0.7208, suggesting that it might be an important tool to assist physicians in clinical practice to treat leprosy. Other datasets need to be used to validate the model and further studies are needed to assess its usefulness in clinical practice.

## Supporting information

**S1 Table. Other registered comorbidities of 329 leprosy patients treated at Souza Araujo outpatient clinic (ASA), Rio de Janeiro, Brazil between 2000–2021.**
(DOCX)

## Author Contributions

**Conceptualization:** Ana Carolina Galvão dos Santos de Araujo, Ximena Illarramendi, Sandra Maria Barbosa Durães, Maurício Lisboa Nobre, Milton Ozório Moraes, Anna Maria Sales, Gilberto Marcelo Sperandio da Silva.

**Data curation:** Ana Carolina Galvão dos Santos de Araujo, Anna Maria Sales, Gilberto Marcelo Sperandio da Silva.

**Formal analysis:** Ana Carolina Galvão dos Santos de Araujo, Mariana de Andrea Vilas-Boas Hacker, Anna Maria Sales, Gilberto Marcelo Sperandio da Silva.

**Investigation:** Ana Carolina Galvão dos Santos de Araujo.

**Methodology:** Ana Carolina Galvão dos Santos de Araujo, Mariana de Andrea Vilas-Boas Hacker, Anna Maria Sales, Gilberto Marcelo Sperandio da Silva.

**Supervision:** Anna Maria Sales, Gilberto Marcelo Sperandio da Silva.

**Writing – original draft:** Ana Carolina Galvão dos Santos de Araujo.

**Writing – review & editing:** Roberta Olmo Pinheiro, Ximena Illarramendi, Sandra Maria Barbosa Durães, Maurício Lisboa Nobre, Anna Maria Sales, Gilberto Marcelo Sperandio da Silva.

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
