## [Decision Letter · Decision Letter 0]

13 Sep 2023

Dear Ms de Araujo,

Thank you very much for submitting your manuscript "Development of a multivariate predictive model for dapsone adverse drug events in people affected by leprosy under standard WHO multidrug therapy" for consideration at PLOS Neglected Tropical Diseases. As with all papers reviewed by the journal, your manuscript was reviewed by members of the editorial board and by several independent reviewers. In light of the reviews (below this email), we would like to invite the resubmission of a significantly-revised version that takes into account the reviewers' comments. 

We cannot make any decision about publication until we have seen the revised manuscript and your response to the reviewers' comments. Your revised manuscript is also likely to be sent to reviewers for further evaluation.

Sincerely,

Alberto Novaes Ramos Jr

Academic Editor

Stuart Blacksell

Section Editor

Reviewer's Responses to Questions

**Key Review Criteria Required for Acceptance?**

**Methods**

-Are the objectives of the study clearly articulated with a clear testable hypothesis stated?

-Is the study design appropriate to address the stated objectives?

-Is the population clearly described and appropriate for the hypothesis being tested?

-Is the sample size sufficient to ensure adequate power to address the hypothesis being tested?

-Were correct statistical analysis used to support conclusions?

-Are there concerns about ethical or regulatory requirements being met?

Reviewer #1: The method section was clear and comprehensive

Reviewer #2: The methods are clearly described.

Reviewer #3: The objective of the study was not clearly defined in the Introduction section. Also, the Introduction section does not contain enough context and justification about the proposed predictive modelling. But, in the Methods section, it was well articulated with a clear testable hypothesis stated. About the study design, I suggest that the authors define clearly, in the Methods section, if is a cohort or a case-control study. Beside this, the report address the stated objective. The population was clearly described and appropriate for the hypothesis being tested. As the authors included an entire cohort, the sample size is sufficient to ensure adequate power to address the hypothesis being tested. Correct statistical analysis were used to support the conclusions and there is sufficient report about ethical or regulatory concerns.

Reviewer #4: This is a nested case/control study, what is most adequate to the objective. The statistical analysis is also coherent to the proposal. The description of the methodological procedures are complete, detailed and acceptable.

**Results**

-Does the analysis presented match the analysis plan?

-Are the results clearly and completely presented?

-Are the figures (Tables, Images) of sufficient quality for clarity?

Reviewer #1: The results were clearly and completely presented and the analysis presented was matched the analysis plan.

All the figures (Tables, Images) have sufficient quality.

Reviewer #2: The results are clearly described. However, it is not clear whether the results are at all surprising or indeed very useful. Table 4 suggests that the only significant predicters are female sex, MB disease and having completed primary education (the last factor may be an artifact, with educated people being more ready to complain about a possible ADE).

Reviewer #3: The analysis presented match the analysis plan. The results were clearly and completely presented. About the figures, I recommend that footnotes are included so that the tables and images are self-explanatory.

Reviewer #4: results are acceptable, well disposed and presented with consistence.

**Conclusions**

-Are the conclusions supported by the data presented?

-Are the limitations of analysis clearly described?

-Do the authors discuss how these data can be helpful to advance our understanding of the topic under study?

-Is public health relevance addressed?

Reviewer #1: Good

Reviewer #2: While the authors describe a nomogram to predeict ADEs in patients taking MDT, they do not develop it into a simple rule-of-thumb that can be used at the bedside. To be useful, it must be simple and quick to use. As mentioned above, this nomogram is of questionable usefulness, as it will simply pick out women with MB leprosy.

A more useful approach to ADEs in relation to MDT, which mainly involve dapsone, would be to suggest proactive steps that health staff may take to prevent some ADEs and identify others at an early and managable stage.

A recent publication on this topic should be referenced: Robin Hilder, Diana Lockwood; The adverse drug effects of dapsone therapy in leprosy: a systematic review; Leprosy Review; 2020; 91; 3; 232-243; DOI: 10.47276/lr.91.3.232

The authors seem happy to emphasize the strength of their study, but they omit to mention any limitations, which include the retrospective design, the lack of many laboratory results and the low practical value of the nomogram they developed.

Reviewer #3: The conclusions are supported by the data presented. The limitations of analysis are clearly described. The authors discuss how these data can be helpful to advance our understandind of the topic under study, but I missed better details on how the proposed predictive model can be useful in clinical or epidemiological practice. I suggest that the authors provide more details on the main topic of the study: the proposed multivariate predictive model.

Reviewer #4: Conclusions are quite straightforward and acceptable.

**Editorial and Data Presentation Modifications?**

Reviewer #1: (No Response)

Reviewer #2: Need for acknowledgement of the weaknesses of the study.

Reviewer #3: (No Response)

Reviewer #4: No need for modifications, apart from Figure 2 (ROC curve) that seems not quite necessary for the general reader and could be removed. Of course, this is a decision of the Editor.

**Summary and General Comments**

Reviewer #1: The manuscript has interesting scope and it contains valuable data. Some modifications are needed:

1. Please recheck all abbreviation and their full name for first time in the manuscript. For examle, line 105 GWAS????

2. Plaese change all "dapsone" to "DDS" in the manuscript, because you used this abbreviation for first time.

3. I suggest you provide an abbreviation list after abstract section

4. Please add the limitations of the study at the end of manuscript.

Reviewer #2: Need for acknowledgement of the weaknesses of the study.

Reviewer #3: I suggest that the authors provide more robust information about the main topic of the study: predictive modelling. In the Introduction section, it is necessary to explore this further, in addition to the issues related to the drug and adverse events. The following can be considered as guiding questions: 1) How can the proposed predictive modelling contribute to clinical and epidemiological practice? 2) Why is it important to propose this predictive modelling? Furthermore, in the last paragraph of the Introduction section, it is important to define the objective of the study: "The objective of this study is to propose predictive modelling..." instead of mentioning that it was done. In the Methods section, it is important to clearly define the type of study: cohort or case-control. The Results section is well reported. In the Discussion section, authors need to indicate how the findings will contribute to clinical and epidemiological practice, as well as give more prominence to the proposed predictive modelling.

Reviewer #4: This is a most welcome study and its results are robust and quite usefeull. Leprosy is a disease with scarce therapeutic choices and the predicion of adverse effects with MDT (the universal acceptable treatment for leprosy) is of utmost importance. If the prediction is made by means of easely identifiable criteria such as social and clinical chracterisitcs, as is the case proposed in this study, the article assumes real importance to improve clinical care of cases. The results can help docotors at any level (including primary health care) to be aware of the possibility of adverse effects and they can mark theses as a special group needing special attention and, for instance, reduing the lag time between clinica revision. In this sense I do reccomend to the editor to consider publishing this artcile.

PLOS authors have the option to publish the peer review history of their article (what does this mean?). If published, this will include your full peer review and any attached files.

Reviewer #1: Yes: Mohammad Sheibani

Reviewer #2: Yes: Paul Saunderson

Reviewer #3: Yes: Gustavo Laine Araújo de Oliveira

Reviewer #4: No
---

## [Decision Letter · Decision Letter 1]

4 Jan 2024

Dear Ms de Araujo,

We are pleased to inform you that your manuscript 'Development of a multivariate predictive model for dapsone adverse drug events in people with leprosy under standard WHO multidrug therapy' has been provisionally accepted for publication in PLOS Neglected Tropical Diseases.

Best regards,

Alberto Novaes Ramos Jr

Academic Editor

Stuart Blacksell

Section Editor

Reviewer's Responses to Questions

**Key Review Criteria Required for Acceptance?**

**Methods**

-Are the objectives of the study clearly articulated with a clear testable hypothesis stated?

-Is the study design appropriate to address the stated objectives?

-Is the population clearly described and appropriate for the hypothesis being tested?

-Is the sample size sufficient to ensure adequate power to address the hypothesis being tested?

-Were correct statistical analysis used to support conclusions?

-Are there concerns about ethical or regulatory requirements being met?

Reviewer #1: Yes

Reviewer #3: The authors included the purpose of the study at the end of the Introduction section, but it is more elegant when it is the last sentence of the section. Furthermore, the authors included context and justification about the proposed predictive modeling. In the Methods section, the authors defined the study design as case-control. This section was well articulated with a clear testable hypothesis stated. The report address the stated objective. The population was clearly described and appropriate for the hypothesis being tested. As the authors included an entire cohort, the sample size is sufficient to ensure adequate power to address the hypothesis being tested. Correct statistical analysis were used to support the conclusions and there is sufficient report about ethical or regulatory concerns.

**Results**

-Does the analysis presented match the analysis plan?

-Are the results clearly and completely presented?

-Are the figures (Tables, Images) of sufficient quality for clarity?

Reviewer #1: Yes

Reviewer #3: The analysis presented match the analysis plan. The results were clearly and completely presented. The figures are of sufficient quality for clarity.

**Conclusions**

-Are the conclusions supported by the data presented?

-Are the limitations of analysis clearly described?

-Do the authors discuss how these data can be helpful to advance our understanding of the topic under study?

-Is public health relevance addressed?

Reviewer #1: Yes

Reviewer #3: The conclusions are supported by the data presented. The limitations of analysis are clearly described. The authors discuss how these data can be helpful to advance our understandind of the topic under study. I noted more details on how the proposed predictive model can be useful in clinical or epidemiological practice.

**Editorial and Data Presentation Modifications?**

Reviewer #1: Yes

Reviewer #3: (No Response)

**Summary and General Comments**

Reviewer #1: A multivariate predictive model for dapsone adverse drug events in people with leprosy were comprehensively suggested.

The manuscript is well written and has clear results and conclusion.

It is proper for publication.

Recommandation: Accept

Reviewer #3: The authors provided more information about the main topic of the study: predictive modeling. In the Introduction section, details about predictive modeling were added. In the Method section, the authors defined the study design. In the Discussion section, the authors briefly addressed how the findings will contribute to clinical practice. In general, the authors addressed the considerations of the first review.

PLOS authors have the option to publish the peer review history of their article (what does this mean?). If published, this will include your full peer review and any attached files.

Reviewer #1: No

Reviewer #3: **Yes: **Gustavo Laine Araújo de Oliveira

---

## [Editor Report · Acceptance letter]

21 Jan 2024

Dear Ms de Araujo,

We are delighted to inform you that your manuscript, "Development of a multivariate predictive model for dapsone adverse drug events in people with leprosy under standard WHO multidrug therapy," has been formally accepted for publication in PLOS Neglected Tropical Diseases.

Best regards,

Shaden Kamhawi

co-Editor-in-Chief

Paul Brindley

co-Editor-in-Chief
